# PeerJ

# Noninvasive analysis of metabolic changes following nutrient input into diverse fish species, as investigated by metabolic and microbial profiling approaches

Taiga Asakura[1,2], Kenji Sakata[1], Seiji Yoshida[1,2], Yasuhiro Date[1,2] and Jun Kikuchi[1,2,3,4]

[1] RIKEN Center for Sustainable Resource Science, Suehirocho, Tsurumi-ku, Yokohama, Kanagawa, Japan
[2] Graduate School of Medical Life Science, Yokohama City University, Suehirocho, Tsurumi-ku, Yokohama, Kanagawa, Japan
[3] Graduate School of Bioagricultural Sciences, Nagoya University, Furo-cho, Chikusa-ku, Nagoya, Aichi, Japan
[4] RIKEN Biomass Engineering Program, Suehirocho, Tsurumi-ku, Yokohama, Kanagawa, Japan

## ABSTRACT

An NMR-based metabolomic approach in aquatic ecosystems is valuable for studying the environmental effects of pharmaceuticals and other chemicals on fish. This technique has also contributed to new information in numerous research areas, such as basic physiology and development, disease, and water pollution. We evaluated the microbial diversity in various fish species collected from Japan's coastal waters using next-generation sequencing, followed by evaluation of the effects of feed type on co-metabolic modulations in fish-microbial symbiotic ecosystems in laboratory-scale experiments. Intestinal bacteria of fish in their natural environment were characterized (using 16S rRNA genes) for trophic level using pyrosequencing and noninvasive sampling procedures developed to study the metabolism of intestinal symbiotic ecosystems in fish reared in their environment. Metabolites in feces were compared, and intestinal contents and feed were annotated based on HSQC and TOCSY using SpinAssign and network analysis. Feces were characterized by species and varied greatly depending on the feeding types. In addition, feces samples demonstrated a response to changes in the time series of feeding. The potential of this approach as a non-invasive inspection technique in aquaculture is suggested.

Corresponding author
Jun Kikuchi, jun.kikuchi@riken.jp

## INTRODUCTION

Most higher organisms such as vertebrates, including humans, harbor a large and diverse number of microorganisms (*Sorokin et al., 2008*) and enjoy significant benefits from such symbiotic relationships (*Backhed et al., 2005*). In symbiotic ecosystems, the microbial symbionts facilitate extensive interactive modulation of metabolism and physiology during

crosstalk with their hosts. Thereby, they can provide traits that the hosts have not evolved on their own, for example, they may synthesize essential amino acids and vitamins or process otherwise indigestible components in the diet (*Gill et al., 2006*). Intestinal microbial symbionts often promote nutritional provisioning and nitrogen recycling (*Douglas, 1998*; *Sabree, Kambhampati & Moran, 2009*) and regulate fat storage for their hosts (*Turnbaugh et al., 2006*). This demonstrates that microbial symbionts have an important role in maintaining metabolic and physiological homeostasis in these ecosystems.

During the investigation of vertebrate intestinal symbionts, mammals such as rodents and humans are commonly studied in terms of microbial structures, diversities, and functions, and metabolic variations and interactions through the crosstalk with hosts; this is despite the fact that mammals comprise fewer than 10% of total vertebrate diversity. Compared to mammals, fish, originated over 600 million years ago, encompass nearly half the total number of vertebrate species with approximately 28,000 extant species (*Nelson, 2006*). Because fish are located near the base of the vertebrate tree of life, investigation of their symbiotic ecosystems are considered important in understanding the co-evolution of mammalian–microbial symbiosis as well as co-metabolic modulations facilitated by the crosstalk between hosts and their microbial symbionts. In this regard, a previous study highlighted a trend of convergent acquisition of similar bacterial communities by fish and mammals, raising the possibility that fish were the first to evolve symbiotic relationships resembling those found among extant gut fermenting mammals (*Sullam et al., 2012*).

When studying co-metabolic modulation by hosts and their microbial symbionts, metabolic profiling is a key approach for characterization and evaluation of metabolism and physiology. Using nuclear magnetic resonance (NMR)-based metabolomic (or metabonomic) approaches provides technical insights for characterizing the interactions of hosts and symbionts (*Brindle et al., 2002*; *Li et al., 2008*; *Nicholson, Lindon & Holmes, 1999*), and this method is capable of generating comparable data between laboratories, thus supporting its continued use (*Viant et al., 2009*). This approach has many advantages for studying host–microbial interactions and assessing metabolic function and homeostasis at the molecular fingerprinting level (*Dumas et al., 2006*; *Nicholson, Lindon & Holmes, 1999*). Therefore, an NMR-based metabolomic approach has been successfully applied to evaluate animal homeostasis, including in human beings (*Claesson et al., 2012*; *Clayton et al., 2006*; *Fukuda et al., 2011*; *Furusawa et al., 2013*; *Holmes et al., 2008*; *Schlipalius et al., 2012*). This NMR-based metabolomic approach is also valuable in aquatic ecosystems for studying the environmental effects of pharmaceuticals and other chemicals on fishes (*Samuelsson et al., 2011*; *Samuelsson et al., 2006*; *Samuelsson & Larsson, 2008*). Such studies have contributed knowledge of basic physiology and development of fish, disease, water pollution, and other aspects (*Bilandzic, Dokic & Sedak, 2011*; *Dove et al., 2012*; *Picone et al., 2011*; *Southam et al., 2008*; *Southam et al., 2011*; *Wagner et al., 2014*; *Williams et al., 2009*). Fish symbiotic ecosystems, however, are rarely investigated, so little information is available concerning metabolic variations associated with fluctuations in microbial composition and structures in diverse fish species affected by the crosstalk between hosts and their microbial symbionts.

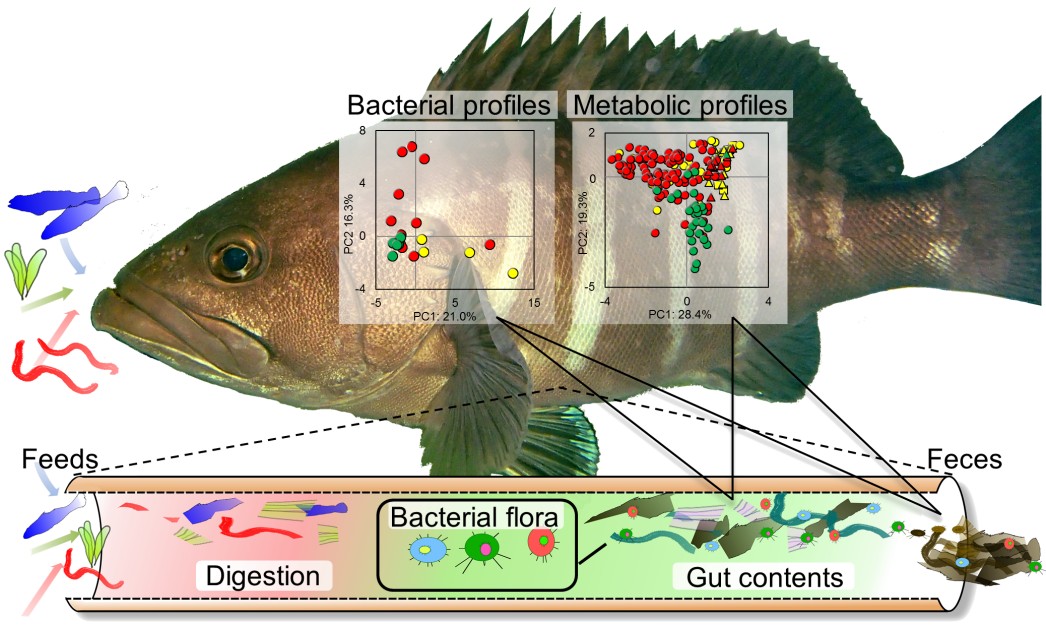

**Figure 1  Experimental overview.** Conceptual diagram illustrating evaluation strategy of metabolic and microbial dynamics in the intestinal environment of fishes using 16S rRNA gene sequence analysis and nuclear magnetic resonance (NMR) with multivariate analysis.

In the current study, we focus on the characterization and evaluation of metabolic variation and diversity associated with the microbial variation and diversity of symbiotic communities in diverse fish species. To this end, the metabolic profiling of fish-microbial symbiotic ecosystems was proposed using a methanol-solvent system for an NMR-based metabolomic approach with the network visualization of the spin coupling of NMR spectra in combination with noninvasive and time-course sampling. Furthermore, we evaluated the microbial diversity in various fish species collected from Japan's coastal waters using next-generation sequencing, followed by evaluation of the effects of different feed types on the co-metabolic modulations in fish-microbial symbiotic ecosystems in laboratory-scale experiments (Fig. 1).

# MATERIALS AND METHODS

## Sample collections and preparations

The fishes used in this study were collected between 2011 and 2013 from estuarine and coastal regions of Kanto, Tohoku, and Southern (Kyusyu-Okinawa) regions. The species of collected fishes, numbers of samples, and their corresponding sampling sites are listed in Table S1 (for sequencing analysis) and Table S2 (for experiments in rearing environment). The fish samples collected were dissected and gutted. The collected gut contents from the whole intestines were lyophilized and crushed (10 min) to a powder using an Automill machine (Tokken, Inc., Chiba, Japan) and used for microbial community analysis.

## Metabolic evaluation of intestinal symbiotic ecosystems in artificially controlled environments

Fish collected from natural environments were reared in laboratory-scale aquariums under artificially controlled conditions. The aquariums were filled about two thirds in volume with artificially synthesized seawater made from "Instant Ocean" (NAPQO, Ltd, Tokyo, Japan), equipped with a Protein skimmer (Kamihata, Inc., Hyogo, Japan), air pump (GEX, Inc., Osaka, Japan), water recycling system (EHEIM, Inc., Chiba, Japan), water cooler (ZENSUI, Inc., Osaka, Japan) and maintained at 18–20 °C. The fish used in this experiment and their feeding conditions are described in Table S2. Feces obtained before providing an artificial diet was labeled as "local feces". The fishes were reared under different conditions thereafter. Feed was given to the fish until satiation 3 days a week. Fecal sampling was implemented every day, and samples were collected unbroken in droppers (SANSYO, Inc., Tokyo, Japan). The collected feces samples were lyophilized and crushed (10 min) to a powder using an Automill machine (Tokken, Inc., Chiba, Japan) and used for metabolomic and microbial analysis.

## NMR measurements

Five milligrams of each of the powdered samples were extracted with 600 μL of deuterated methanol solvent with 1 mM of sodium 2,2-dimethyl-2-silapentane-5-sulfonate (DSS) as an internal standard at 60 °C for 15 min. After centrifugation at 25 °C for 5 min, the extracted supernatant was transferred to a 5 mm NMR tube for the NMR measurements. One-dimensional (1D) spectra of the samples were acquired at 298 K using a Bruker AVANCE II 700 spectrometer equipped with an $^1$H inverse triple resonance cryogenically cooled probe with $Z$-axis gradients (Bruker BioSpin GmbH, Rheinstetten, Germany), as described previously (*Date et al., 2014*). In brief, $^1$H NMR spectra were acquired by the standard Bruker pulse program p3919gp, with 32 K data points with a spectral width of 14 ppm, were collected into 32 transients and 8 dummy scans. Prior to Fourier transformation, free induction decays were multiplied by an exponential window function corresponding to a 0.3 Hz line broadening factor. The acquired spectra were manually phased and baseline corrected.

The method for the NMR measurement of the two-dimensional $^1$H–$^{13}$C heteronuclear single quantum coherence (HSQC) and total correlation spectroscopy (TOCSY) has been previously described (*Sekiyama, Chikayama & Kikuchi, 2010*; *Sekiyama, Chikayama & Kikuchi, 2011*). In the HSQC NMR spectra, a total of 128 complex F1 ($^{13}$C) and 1,024 complex F2 ($^1$H) points were recorded from 256 scans per f1 increment. The spectral widths were 150 and 14 ppm for F1 and F2, respectively. The TOCSY spectra were acquired in the range of 10.7 to −1.7 ppm using 4096 (F2) and 512 (F1) data points with 16 scans and an interscan delay of 2 s with 16 dummy scans. The mixing time (D9) was set to 90 ms. The HSQC-TOCSY spectra were acquired as 1,024 (F2) and 256 (F1) data points with 180 scans and an interscan delay of 1 s with 16 dummy scans. The NMR spectra were processed using Bruker TopSpin software (Bruker BioSpin GmbH) and assigned using the SpinAssign program at the PRIMe website (http://prime.psc.riken.jp/) (*Akiyama et al., 2008*;

*Chikayama et al., 2010*; *Chikayama et al., 2008*) and the Biological Magnetic Resonance Bank (http://www.bmrb.wisc.edu/metabolomics/query_metab.php) (*Ulrich et al., 2008*).

## Microbial community analysis

Microbial DNA extraction was performed according to previous studies with slight modifications (*Date et al., 2012*). Each DNA sample was amplified by PCR using universal bacterial primers 954f and 1369r targeted to the V6–V8 regions of the 16S rRNA gene according to a previous report (*Date et al., 2010*). The sequencing analysis and the data processing was outsourced to Operon Biotechnologies Co. Ltd. (Tokyo, Japan). The categorizations of bacterial taxa were performed using a Ribosomal Database Project (RDP; http://rdp.cme.msu.edu/seqmatch/seqmatch_intro.jsp) classifier (*Wang et al., 2007*).

## Statistical analysis

All 1D $^1$H-NMR data were reduced by subdividing the spectra into designated sequential 0.01-ppm regions between $^1$H chemical shifts of 0–9 ppm. After the exclusion of water resonance, each spectrum was normalized by constant sum. Principal component analysis (PCA) was run on R software and performed according to a previous report (*Date, Sakata & Kikuchi, 2012*). The data were visualized in the form of the PCA score plots and loading plots. Each data point on the score plots represented an individual sample, and each data point on the loading plots represented one NMR spectral data point related to the metabolites. Partial Least Squares Discriminant Analysis (PLS-DA) was performed on R software with a package "muma". The network diagram was rendered in Gephi (http://gephi.org) according to previous studies (*Ito et al., 2014*; *Yamazawa et al., 2014*).

# RESULTS AND DISCUSSION

## Microbial diversity in gut symbiotic ecosystems of various fish species

This study focused on microbiota and metabolic variation of gut contents in various fish species living in estuarine and coastal regions of Japan. To evaluate the diversity of gut microbiota, the microbial composition and structure of the gut contents of various fish species listed in Table S1 were analyzed using a next-generation sequencer. Almost all fishes had common bacterial Phyla such as Phylum Proteobacteria, Phylum Firmicutes, and Phylum Actinobacteria as dominant bacterial categories with some exceptions (Fig. S1). In order to compare the microbiota profiles among fishes, PCA was performed on the sequencing data (Figs. S2 and S3). From the PCAs, omnivorous fish were likely to be located on the negative PC1 side, whereas carnivorous fish were likely to be located on the positive PC1 side. The analysis of diversity index on the basis of the Shannon index showed that the trend for omnivorous fish was toward high diversity and was located on the negative PC2 side, whereas the profiles with relatively low diversity index values were located on the positive PC2 side (Fig. S2A). Diadromous fish tended to be clustered on the negative PC4 side (Fig. S2B). Based on the loading plot analyses, the factors contributing to omnivorous feeding habitat clusters were shown as, e.g., unclassified Hyphomicrobiaceae bacterium, *Mycobacterium* sp., and *Ilumatobacter* sp., whereas the factors contributing to

carnivorous feeding habitat clusters were shown as, e.g., *Methylobacterium* sp., *Bacillus* sp., and *Arthrobacter* sp. (Fig. S2C). The contributing factors for the fishes showing low diversity index values in their guts were shown as, e.g., unclassified Vibrionaceae bacterium and *Clostridium* sp., whereas the contributing factors for diadromous fish were shown as, e.g., *Deinococcus* sp. and unclassified Micrococcaceae bacterium (Figs. S2D and S2E). Fish with differing feeding habits have characteristic microbiota; therefore, it is possible to characterize the fish according to their intestinal microbiota. These changes were expected to affect the physiology and health of the fishes, and therefore we performed the metabolomic evaluation using a diverse range of fishes selected from the above species.

## Metabolic characterization of gut content and feces using methanol solvent system

In aquafarming industries, non-invasive sampling to evaluate the health and physiological effects of the environment on fish should be possible and would contribute significantly to the farming process. Thus, we focused on metabolic profiling of fish feces for evaluating health and physiological variation. For this purpose, the water extraction method for NMR measurements is important when similar metabolomics are performed on humans and mice, however, because sampling in water is unavoidable with fish, this caused concern that many water-soluble components may be diffusing into the water. Therefore, in the present study, we analyzed the fractions obtained using methanol solvent, and the metabolic variability of the fractions was evaluated by $^1$H NMR spectra of feces of 14 fish (Fig. S4). For the annotation of metabolites extracted from methanol solvents, we measured mixtures of three diets (Polychaeta, Crustacea, and Aquaculture feeds), feces, and intestinal contents of *Epinephelus septemfasciatus* and used our originally customized database (SpinAssign program) for HSQC annotations with the network visualization of the spin coupling on TOCSY and HSQC-TOCSY NMR spectra to enhance the annotation of metabolites (Figs. 2, 3, S5 and S6 and Table S3). However, the methanol solvent also extracts polar metabolites that are water-soluble, and therefore, the retention capability of water-soluble components in feces was evaluated by a water rinse experiment for fish feces. The collected feces were rinsed three times with artificial seawater for 15 min with shaking. The fecal samples before and after rinsing were extracted with methanol and measured by NMR spectroscopy. Almost all signals excluding leucine detected in HSQC NMR spectra were retained after rinsing, as shown in Fig. S6. In particular, water-soluble components such as amino acids and sugars were detected in feces by HSQC spectra using methanol extraction (Figs. S5 and S6). The retention of water-soluble components may be because of a physical property of feces, i.e., the water-soluble components may be packed and aggregated in feces either chemically (by chemical and physical interactions) and/or biologically (by biological interactions such as retention in cells and/or trapping by microbes).

## Comparisons of microbial and metabolic profiles among diets, intestinal contents, and feces of fish

To evaluate the microbiota and metabolic profiles of gut contents and feces in a rearing environment, sequencing analysis and $^1$H NMR-based metabolic profiling were performed

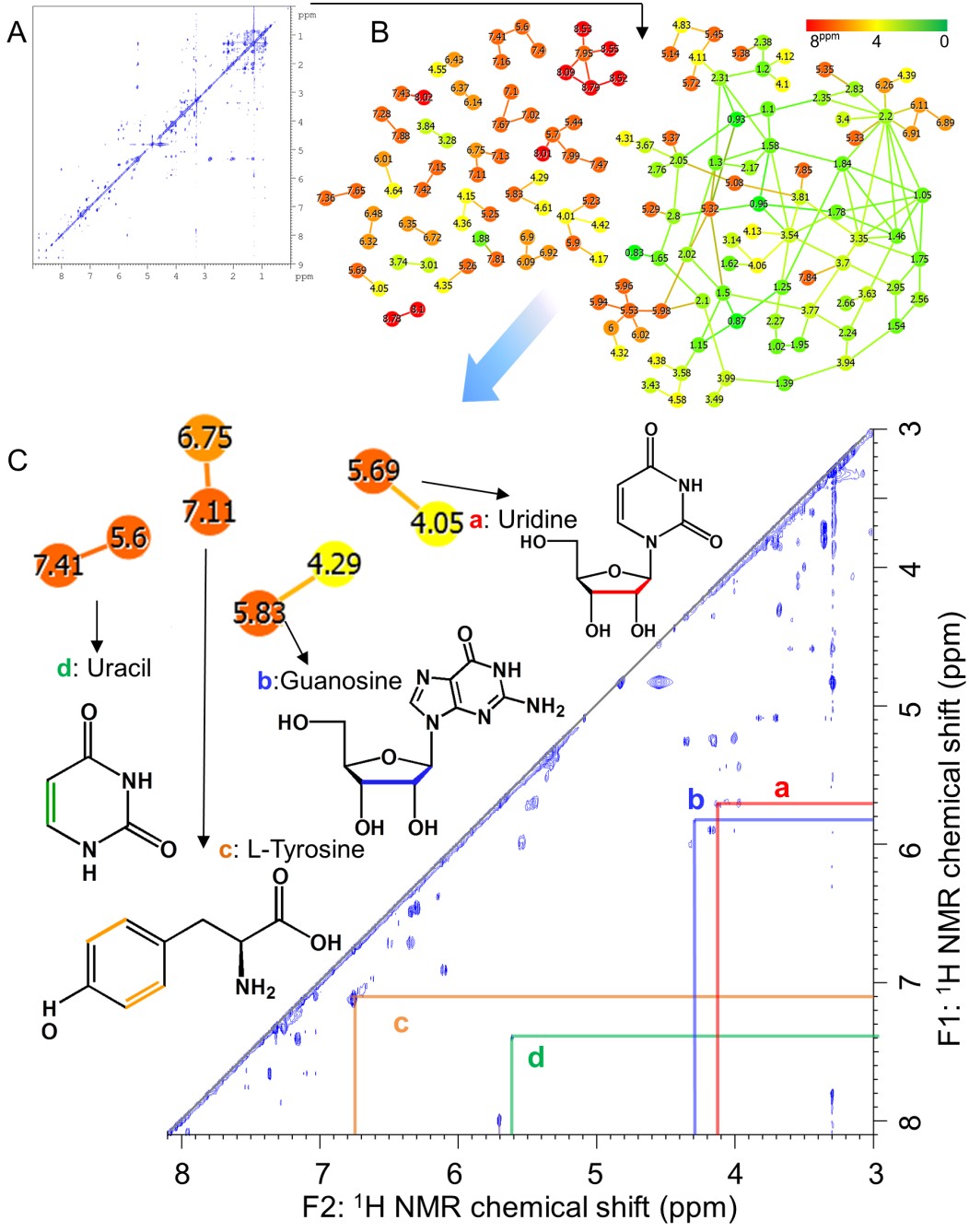

**Figure 2 Advanced analytical approach using a network visualization method based on TOCSY spectra.** (A) Representative TOCSY NMR spectra for methanol-soluble components in fish feces. (B) Visualization of spin coupling network of TOCSY spectra based on correlation network analysis. Green indicates a low magnetic field and red indicates a high magnetic field in $^1$H NMR chemical shift. The nodes are shown as individual $^1$H NMR spectral peaks, and lines are drawn on the basis of the cross-peaks of TOCSY spectra. (C) Annotated peaks by steps A and B.

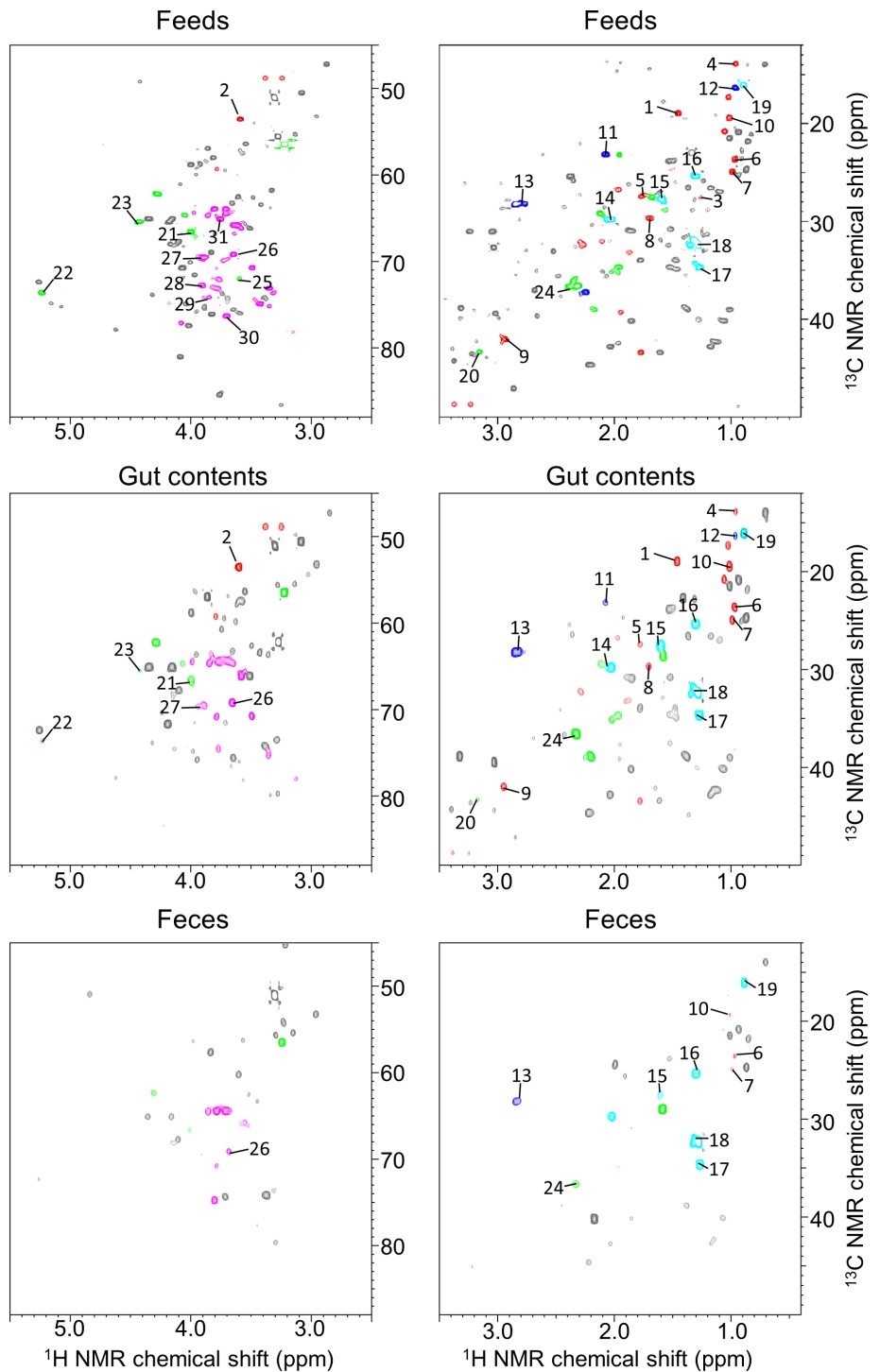

**Figure 3** **¹H–¹³C HSQC spectra of methanol-soluble component from fish feeds, intestinal contents, and feces.** Spectral intensities were normalized to an internal standard (DSS). Red, amino acids; blue, unsaturated fatty acids; aqua, fatty acids and phospholipids; green, phospholipids; pink, sugars.

in combination with evaluation by PCA (Fig. S7). The microbiota and metabolic profiles in feces showed trends of clustering on the basis of feeding conditions, whereas those profiles in gut contents showed no clear clustering under the feeding conditions. This result indicated that the feces as an aggregate of final metabolic products were influenced by feedings compared with gut contents. It also suggested covariations between the microbial community and metabolites in feces and gut contents on the basis of the differences in feeding conditions. In addition, on the basis of the intensities of the annotated peaks evaluated by HSQC and TOCSY spectra using the network visualization approach, we further analyzed and compared the metabolic profiles among diets, intestinal contents, and feces of fish (Fig. 4). The concentrations of these samples were uniform in 50 mg of feces/600 µl of MeOD and were normalized by the internal standard (1 mM DSS). They were compared in relative amounts on the basis of the peak of feeds (as 1.0) in Fig. 3. Sugar, lipid, and peptide peaks were commonly observed in data from the feeds. A relatively high number of lipid peaks were observed in feces samples and intestinal contents, whereas the peaks from sugars and peptides were reduced. Many peaks detected in the intestinal contents were no longer present in the feces samples. Based on the peak intensity of the individual metabolites, reduction rates were evaluated for the intestinal contents and the fish feces for each feed type. The levels of evaluated metabolites decreased in the change from intestinal contents to feces for each feed type and were lowest in the feces. The reduction rate was characterized by the respective substances. Assuming that feeds are 100%, the phospholipids and sugars were reduced to less than 30% by the intestinal content stage, and had become approximately 10% in the feces. In contrast, the lipid peak was more than 40% in the intestinal contents and 20% in the feces. It was shown that absorption rates of each metabolite had a characteristic profile. Further, the metabolites that were difficult to decompose remained in the intestines and were lost in the feces; they were considered to be incompletely absorbed by host fish species and unavailable by intestinal microbiota. From these results we can see that metabolic information about the fishes was reflected in the methanol fraction of the feces.

## Comparisons of metabolic diversity of various fish species under artificially controlled environments

To evaluate the metabolic dynamics in intestinal content in various fish species and to characterize the effects of a change in diet on fish-microbial symbiotic ecosystems over time course variations, we focused on the effects of different feed types on the co-metabolic modulations in fish-microbial symbiotic ecosystems in laboratory-scale experiments. For this experiment, 18 species of fish were selected including *Acanthogobius flavimanus*, *Sebastes ventricosus*, *Sebastiscus marmoratus*, and *Sillago japonica*; the rearing conditions for each fish are listed in Table S2. The different metabolic profiles depending on the fish species were obtained by [1]H NMR measurements (Fig. S4). In the evaluation of PCA, the metabolic profiles of the 18 fish species appeared to be largely changed by feeding situation rather than by fish species (Fig. 5A). By loading plot analyses, some metabolites such as fatty acids and phospholipids contributed to the clustering of aquaculture feeds (Fig. 5B). In feces from the rearing environment of *Acanthogobius flavimanus*, for example,

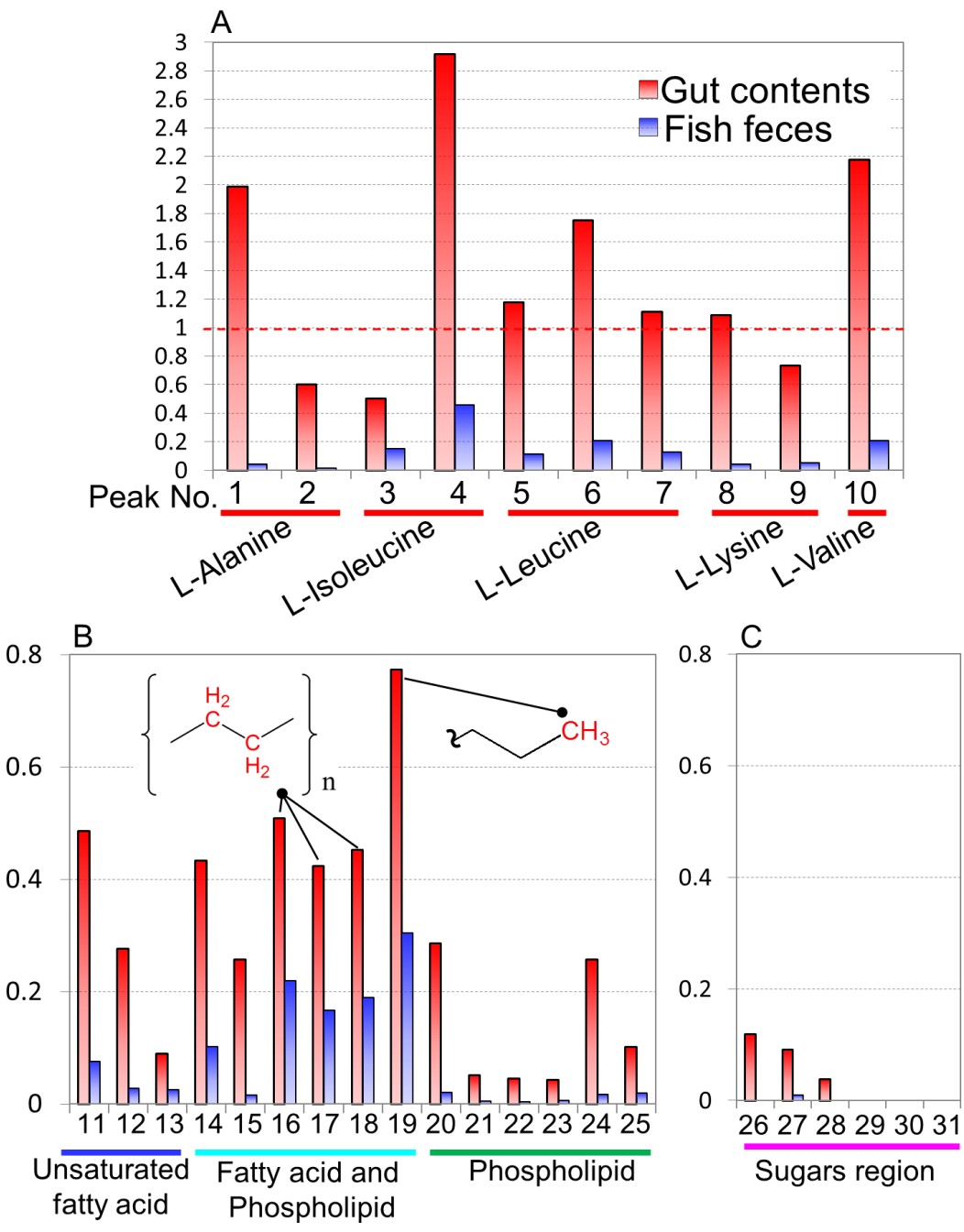

**Figure 4 Comparisons of spectral intensities in each detected metabolite among *Epinephelus septem-fasciatus* feces, intestinal contents, and feeds.** Relative peak intensities of the feces and intestinal contents from fish on the basis of the intensities of feeds are shown. Peak numbers and annotated metabolites are listed in Fig. S5 and Table S2. (A) amino acid, (B) unsaturated fatty acid, fatty acid, and phospholipid, (C) sugar region peaks.

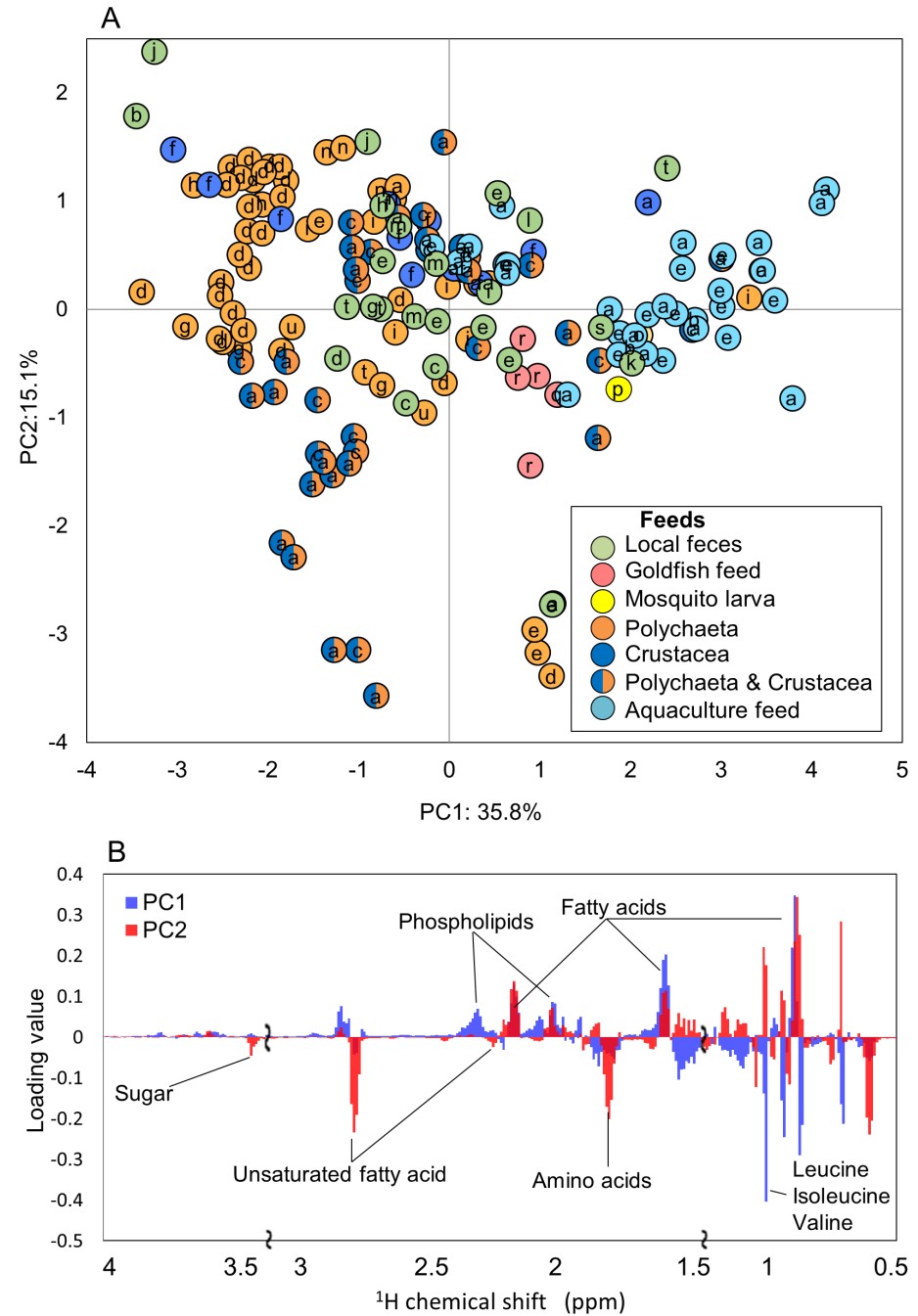

**Figure 5 Metabolic variations based on ¹H NMR profiling of various fishes evaluated by PCA.** (A) PCA score plot for ¹H NMR profiles ($k = 793$, $R^2X = 0.268$, $R^2Y = 0.181$, $Q^2 = 0.201$) of fish feces was computed from 169 samples from 21 species. Letters (a–u) represent fishes listed in Table S2. (B) PCA loading plots on ¹H NMR profiles.

phospholipid, fatty acid, and trimethylamine were increased. Local (natural environment) feces were diffused near the center; it can be seen that the diversity was greater than that in the breeding environment. In addition, there were trends, such as phospholipids and amino acids in local feces were lower than those of the breeding environment from the loading plot analysis. This trend was speculated to result from differences in the composition and digestibilities of protein and lipids in live food and in artificial diets. Profiling of fish reared on the same feed was strongly clustered, and there was a tendency to cluster by species. In addition, the digestion and absorption of feed in different species was reflected in their feces.

Time-series behavior based on differences of feeding in the same species, were evaluated by profiling methods described above in the rearing environment. Results of PCA show that feces profiles varied greatly but temporarily due to a change in feed and tended to return to the original profile after a period of time (Fig. 6A). Based on the loading plot analysis, some metabolites such as phospholipids and fatty acids contributed to the variations by the impact of aquaculture feeding (Fig. 6B). In addition, microbiota profiles widely varied with the effect of feeding change by aquaculture feed, but the community was likely to be restored when carnivorous feeding was again available to the fishes (Fig. 6C). This result suggests that feeding changes in *Epinephelus septemfasciatus* affected the variations in metabolic and microbiota profiles in feces. In these results, a difference in response to time series of feeding associated with fecal metabolic and microbiota profiles can be observed. With information of the intestinal environment being reflected in the fish feces, we propose that this could be an informative non-invasive technique for cultured fish.

In this study, we evaluated the metabolic and microbiota profiles in fish-microbial symbiotic ecosystems. To evaluate variations in metabolism of intestinal symbiotic ecosystems using noninvasive sampling, one of the disadvantages was the water-solubility of components of the fish feces due to their aquatic environment. To circumvent this issue, we applied the methanol-solvent-based metabolomic approach with a proposal that metabolic and microbiota profiling in fish feces using the methanol solvent system could be an informative noninvasive technique for cultured fish. The application of our approach is a step toward clarifying the metabolic dynamics of the complex intestinal microbial community living inside fish in aquatic environments. It also translates microbial activity and function into host responses, and may even provide a means to engineer the metabolic activities of intestinal microorganisms to improve the health of fish in the form of prebiotic or probiotic treatments in the aquafarming industry.

In summary, this study demonstrates the metabolomic analysis of fish feces with noninvasive sampling using methanol-solvent systems with the network visualization approach based on TOCSY spectra in combination with HSQC peak lists annotated by the SpinAssign program. Using this approach, we evaluated the metabolic diversity of various fish species from coastal environments around Japan while they were in laboratory-scale rearing systems and revealed that the diversity was caused by differences in the feeding situations rather than the diversity of fish species.

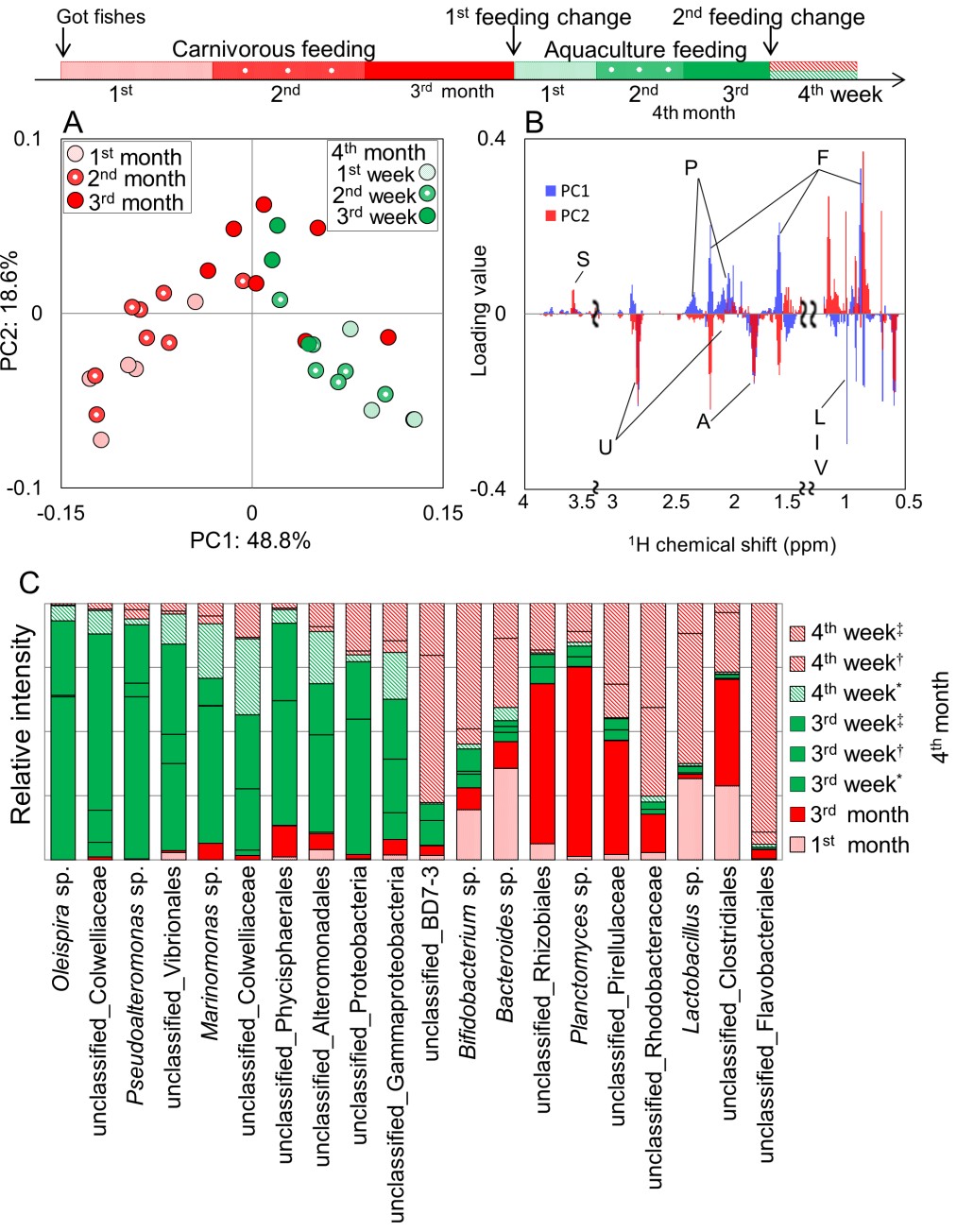

**Figure 6 Metabolic and microbial profiles of feeding response in *Epinephelus septemfasciatus* feces.** Symbols representing individual communities are colored by diet (red, carnivorous feeding; green, aquaculture feeding). Three *Epinephelus septemfasciatus* were bred by carnivorous feeding for 3 months. They were then divided into three tanks (*, †, ‡), and bred for 3 weeks by aquaculture feeding. One *E. septemfasciatus* was then bred with aquaculture feeding, and the others were bred with carnivorous feeding. (A) PCA score plot on $^1$H NMR profile from feces ($n = 38$, $k = 808$, $R^2X = 0.488$, $R^2Y = 0.187$, $Q^2 = 0.442$). (B) PCA loading plots on $^1$H NMR profiles. Letters S, U, P, A, F, L, I, and V indicate sugar, unsaturated fatty acid, phospholipids, amino acids, fatty acids, leucine, isoleucine, and valine, respectively. (C) Bacterial profile that changed characteristically during the 4 months of rearing. These data were first normalized by total reads for each sample (i.e., the ratio in each fecal sample), and then secondarily normalized on the basis of the sum of the ratios of the same bacteria.

## ACKNOWLEDGEMENTS

The authors wish to thank Yuuri Tsuboi, Amiu Shino (RIKEN), and Yuho Sato (Yokohama City University) for stimulating discussions, technical assistance, and useful advice on NMR measurements and analysis.

### Funding

This research was supported in part by Grants-in-Aid for Scientific Research (Grant No. 25513012) (to JK), the Advanced Low Carbon Technology Research and Developmental Program (Grant No. 200210023, ALCA to JK) from the Ministry of Education, Culture and Sports, and RIKEN Junior Research Associate Program. The funders had no role in study design, data collection and analysis, decision to publish, or preparation of the manuscript.

### Grant Disclosures

The following grant information was disclosed by the authors:
Grants-in-Aid for Scientific Research: 25513012.
Advanced Low Carbon Technology Research and Developmental Program: 200210023.
Ministry of Education, Culture and Sports, and RIKEN Junior Research Associate Program.

### Competing Interests

Kenji Sakata, Yasuhiro Date and Jun Kikuchi are employees of the RIKEN Center for Sustainable Resource Science (CSRS).

### Author Contributions

- Taiga Asakura performed the experiments, analyzed the data, wrote the paper, prepared figures and/or tables, reviewed drafts of the paper.
- Kenji Sakata and Seiji Yoshida performed the experiments.
- Yasuhiro Date analyzed the data, wrote the paper, prepared figures and/or tables, reviewed drafts of the paper.
- Jun Kikuchi conceived and designed the experiments, contributed reagents/materials/analysis tools, wrote the paper, reviewed drafts of the paper.

### Animal Ethics

The following information was supplied relating to ethical approvals (i.e., approving body and any reference numbers):

There is no specific permission required for fish study in Japan.

### Field Study Permissions

The following information was supplied relating to field study approvals (i.e., approving body and any reference numbers):

There is no specific permission required for all of the sampling points as they are public places. Also the field does not host endangered or protected species.

## Supplemental Information

Supplemental information for this article can be found online at http://dx.doi.org/10.7717/peerj.550#supplemental-information.

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
