# Peer review of "Noninvasive analysis of metabolic changes following nutrient input into diverse fish species, as investigated by metabolic and microbial profiling approaches"

_PeerJ, doi:10.7717/peerj.550_

## Round 0.1 · original submission · Major Revisions

Your MS has been reviewed by two independent referees. Both raise a number of points which all need to be taken into account.

Reviewer 1 ·

Basic reporting

The manuscript by Asakura et al. describes the non-invasive analysis of metabolic changes in fish using NMR-based metabolomics and pyrosequencing. This could potentially be an important contribution to the field, and is definitely of interest to the readers of PeerJ.

However, in the present form there are several issues wiht the manuscript that should be corrected before it is suitable for publication in PeerJ:

1. The Figure legends are rather short, and omit important imformation, e.g:
1.1. Fig 5: What do the blue and red cylinders in the top left panel mean?
1.2. Fig 5: How were the %ages of spectral intensity in feces or gut contents relative to feed calibrated? Was the gram (or better molar!) amount of feces and feed used for the NMR spectra similar? Just dividing raw spectral intensities from 2 NMR spectra by each other is meaningless if the amount of maerial is not similar. Please describe.
1.3. Fig 6: I assume "numbers (1-18)" mean "Letters (a-u)" ?
1.4. Fig 6: What does the colour scale in the loadings plot mean? (different blue/red shades?) Similarly Fig S4B.
1.5. Fig S3: "1H-NMR spectra of methanol-soluble [WHAT??] from 17 fish feces (1) to (14). (1) to (14) fish are listed in Table S2."
1.5.1. Describe the missing "What?" part of the sentence.
1.5.2. The (1)-(14) fish are not described in Table S2 (or for that matter any other table). Please correct the legend, and provide this description of the fish in adequate form.
1.6. Please amend the legends of the other figures to provide sufficient and adequate description of the figure contents.

2. Minor grammar errors should be corrected, e.g
2.1. delete "the" in line 110.
2.2 Fig S4(a) It is "1st", "2nd" and "3rd", not "1st", "2th" and "3th".

Experimental design

While the experimental design seems to be basically solid, the technical description of the design and experiments is not up to standard, and thus it is not entirely possible to judge the design and execution of experiment and data analysis (or for that matter repeat them). This should be amended.

Specific points:

3. How many fish were used from each species to conduct the study? Surely there were more than one fish per species and collection point or feeding regime? These numbers should be added as addiitonal columns to Tables S1 and S3.

4. Methods: The description of the 1D NMR methods (lines 112-119) is incomplete. It is not mentioned whther 1H or 13C spectra were collected. Judging from the spectral width of 150 ppm 13C spectra were collected, but the paper shows only 1H 1D spectra. Should this be "14 ppm"? Please describe the experiment/pulseprogram used to aquire the spectra.

5. Statistical analysis: This description is also not according to the minimum standards recommended by the International Metabolomics Society:
5.1. Spectra were "normalized by an internal standard and constant sum". How exactly does this work? To my understanding both methods are mutually exclusive in that normalising to "constant sum" normalises each spectrum to a total intensity of 1, while "normalising to an internal standard" assigns different intesities to each spectrum. What method was used and exactly how?
5.2. line 145: Replace each occurrence of "coordinate" by "data point" in this sentence.
5.3. The description of the PCA and PLS-DA models is sub-par. Eg. how many samples (n) are in each model? How many variables (k)? What scaling was applied to the variables? What Y-data were used in the PLS-DA, and how were they structured and scaled? How many principal/latent components (A)? How many % of the variation are explained in total and per component by the model (R2)? What is it's predictability (Q2)? These are standard figures of merit for such anayses that should be included either in a suplementary table or be included in the legends of the figures where the respective models are presented.

6. The data on microbial communities in the fish and the metabolmic analyses stand in the paper next to each other, but are not connected to each other. The microbial data are mentioned in Figs 2, S1 and S2, but there is no analysis on whether how there are any correlations/connections between gut microbiota and metabolite profiles in either gut content or feces. This is a missed opportunity of this manuscript, and it is at present not quite clear what the pyrosequencing data add to the manuscript apart from additional (and at present unrelated) information.

7. The authors mention that one problem with sampling fish feces from an aquatic environment is that water-soluble metabolites could have been lost from the feces into the aquarium/open sea prior to sampling. They try solving this by using pure methanol for fecal metabolite extraction. However, they fail to discuss whether that is really a solution, as methanol also extracts mainly the same polar metabolites that are also water-soluble and thus in danger of being lost. There would be a minor fraction of metanol-soluble but water unsoluble metabolites which sould be extracred, but for the majority of metabolites detected, the same problem of loss into water still applies. Is this really a problem? As long as the loss is either consistenly the same across all samples, or at least not systematically different between samples, this should be more of an inconvenience adding statistical noise, rather than a problem requring a new solvent system. If the authors are after the the samll fraction of water-insoluble but methanol-soluble metabolites then this should be stated in the paper.

Validity of the findings

The data analysis is not described in sufficient detail to ascertain the validity of the data or some of the findings. This should be improved.

8. lines 158-160: "The gut microbiota profiles of the fish species were expected to form clusters on the PCA score plots according to the regions from which they were collected and their feeding habitats (Fig. S2)."
When looking at Fig S2, the data do not form any visible clusters, neither based on region nor on feeding habitat. But the fact that the data do not conform to the (reasonable) expectation is not discussed anywhere in the paper. Please discuss!

9. When looking at the data for microbiota profiles, there is no clustering according to location or feeding regime in the unsupervised PCA in Fig S2 (see also point 8). However in the *supervised* PLS-DA of the *same* data there is clustering according to region (Fig 2A) or feeding regime (Fig 2C). The fact of getting separation according to *two* different criteria in the PLS-DA, when there was no separation in a PCA, is strongly raising the suspiction that the PLS-DA models are overfitted. (Using PLS-DA it is possible to generate perfect separation between individual groups when using completely random and non-sensical data as input!)
9.1. Please describe how the PLS-DA models were constructed and especially which structure the Y-data had (see also point 5.3).
9.2. Were PLS-DA models constructed that investigated separation according to region and feeding regime at the *same* time in e.g. a two-column Y-table? How do these models compare to the (I assume 1-column Y-table) models presented in Fig 2?
9.3. Please validate all PLS-DA models presented by permutation analysis (permuting the identity labels in the Y-table eg. 200x) and show the results of the validations as plots in the supplementary material. This is absolutely critical to evaluate the validity of the presented models.

10. Fig S4(a): I find it curious that when looking at the metabolic trajectory of Ephinephelus septemfasciatus (please italicise species name in Figure legend), the follwing patter is observed: (1) under polychaeta/crustacea diet the mtabolite profile changes continuously during the first 3 monts of the diet. (2) when aquaculture feed is introduced there is a sudden change in metabolite profile (this would be expected). (3) within a month on aquaculture feed the metabolite profile chanfges back to a state that is virtually indistinguishable from the final metabolite profiles on polychaeta/crustacea feed. Do the authors have any idea or speculation on what is going on here? Are the sample points at 3rd month and at the 3rd week of month 4 separated in the third dimension of the PCA (PC3)? What is going on here biologically? Or is this a labelling error, and the weeks 1-3 of month 4 should be inverted in the legend (thus creating a continuous metabolic trajectory)?

11. The authors describe the problems of assigning NMR signals in pure methanol to individual metabolites, and focus on the described network-based visualisation approac as "a powerful and useful tool". However juding by the contents of Table S2, this approach seems to deliver raather ppor results: The majority of the peaks in the HSQC spectra is unannotated and unidentified, and for the peaks that are annotated there are multiple (often more than a dozen) possible candidates. There is no evidence that this candidate list has been narrowed down to individual unambiguously identified metabolites by the TOCSY-based coupling networks. Is this really the final stage of metabolite identification? If so, the method is not as powerful as described. Please discuss this point in more detail int he manuscript, and if the metabolite identifiaction is indeed better than what is currently presented in the paper, please update the information to the level of quality that was finally achieved.

12. Please deposit acquired NMR spectra in an approriate public data base, such as MetaboLights. The manuscript currently does not contain information that this has been done.

Reviewer 2 ·

Basic reporting

The work presented by Asakura T and co-authors on « Non-invasive analysis of metabolic changes following nutrient input into diverse fish species, as investigated by metabolic and microbial profiling approaches » is innovative and original. The authors implement relevant approaches for describing and characterising the diversity of gut micro-organisms and metabolites in fish. There are however strong weakness in the manuscript. It could thus be published in Peer J but with major modifications.

The work proposed two very separate parts : the diversity of microbial community in the gut related to feeding habits and the effect of diet on metabolite profiles of intestine and feces.
• There is no attempt to connect these two parts. The differences observed in metabolite profiles of gut are never related to the microbial diversity. This require a further analysis of the data.
• Otherwise the authors have to split this work in two distinct papers. If a choice has to be made the originality of this work is mainly the characterisation by NMR approaches of metabolites profiles in gut and feces of various fish species. The approaches implemented to identify metabolites is relevant. The effect of diet is also original (supplemental figure 4).

The authors use some terms which are not relevant to the processes studied.
• The use of metabolism (title : « metabolic changes following nutrient input » and in the text « metabolism in fish » or « co metabolic processes ») is confusing. This work is not related to fish metabolism.
• This work concerns two different processes : digestion/absorption on the one hand and microbial metabolism on the other hand. Other analysis of the data have also to be realised to illustrate the differences related to each of the two processes. For instance, the quantitative differences observed between diet, gut and feces are clearly related to digestion/absorption process. The authors have to illustrate differences specifically related to microbial metabolism.

The summary, introduction and conclusion has to be corrected by an english native reader.
Figure 4 : A and B are mentioned in the legend but are not in the figure
Figure 6 : replace in the legend Numbers (1-18) by Letters (a to t). The color range used is not adapted to discriminate visually the different feeds.

Experimental design

The interest of studying metabolites profiling in the feces as an non invasive approach of the diversity and the quality of the diet, is not discussed further.
• The main difficulty of quantifying remaining compounds in feces is related to the leaching of feces.
• This has been already been studied for macro-molecules protein, lipids, but also minerals and vitamins. It depends on species (different physical structure of feces), diet composition and water current. It is thus non predictible. For this reason the continuous sampling of feces or the stripping of fish to collect newly formed feces is currently practice.
• Furthermore the soluble compounds are especially affected by the leaching of feces. The quantitative assessment of these compounds such as the metabolites studied in this work is thus not feasible without a specific protocol for collecting the feces.
• It could be supposed however that the leaching process affect equally all the compounds. Thus the remaining compounds could be representative qualitatively.
• The use of a methanolic extraction of feces is proposed in this work. The representativity of this extract could only be assessed from the proximity between metabolic profile of gut and feces (figure 5). Furthermore methanol extraction could extract compounds that are trapped in the feces matrix and that are not absorbable (see comments below). This could be analysed and discussed further.

Validity of the findings

The authors seems to ignore the basis of nutrition
• The gut content represent a complex mixture between the contents of the stomach, the upper intestine, the pyloric caeca in some species and the lower intestine. The contents of all these parts are at different stages of digestion / absorption processes. It is thus important to specify if gut content have been collected on whole digestive tract or sampled in a given part of the digestive tract or not.
• l 212–216 The calculation of a reduction rate for individual metabolites between diet and gut is not relevant except for few specific molecules generally not considered as metabolites for fish. Due to degradation of macro-molecules (protein, lipid) by digestive enzymes the metabolites detected in the diet are massively diluted (at least a 100 factor) by those released from the digestive processes. It is thus very surprising that lipids, phospholipids and even sugars decreased from 100 to 30 – 40 % between diet and gut when they are supposed to increase.
• l 218-220 The metabolites still remaining in the feces are considered by the authors as « unabsorbable or not degradable by microorganism ». The metabolites of lipids, amino acids and sugars identified in the feces are really absorbable. Thus the remains analysed in the feces means the absorbtion process is not complete. Free aminoacids and sugar are not supposed to be degraded by microorganisms as fermentation process did not occur in fish digestive tract.
• l 237-240 The authors interpreted the differences in phospholipids and amino acids content in the feces between wild and breeding condition in the light of differences in N and P nutrients availability in the wild environment as compared to artificial diet. This is a non sense, this is rather resulting of differences in the composition and the digestibilities of protein and lipid in live food and in artificial diet.

Additional comments

No comments

---

## Round 0.2 · accepted · Accept

We see that you have addressed all the different points raised by the reviewers.